# Results of the Hungarian Newborn Screening Pilot Program for Spinal Muscular Atrophy

**DOI:** 10.3390/ijns11020029

**Published:** 2025-04-23

**Authors:** Krisztina Hegedűs, István Lénárt, Andrea Xue, Péter Béla Monostori, Ákos Baráth, Borbála Mikos, Szabolcs Udvari, Adrienn Géresi, Attila József Szabó, Csaba Bereczki, Mária Judit Molnár, Ildikó Szatmári

**Affiliations:** 1Paediatric Centre, Semmelweis University, 1083 Budapest, Hungary; hegedus.krisztina1@semmelweis.hu (K.H.); xue.andrea@semmelweis.hu (A.X.); szabo.attila@semmelweis.hu (A.J.S.); 2Department of Paediatrics, University of Szeged, 6725 Szeged, Hungary; lenart.istvan@med.u-szeged.hu (I.L.); monostori.peter.bela@med.u-szeged.hu (P.B.M.); barath.akos@med.u-szeged.hu (Á.B.); bereczki.csaba@med.u-szeged.hu (C.B.); 3Bethesda Children’s Hospital, 1146 Budapest, Hungary; mikos.borbala@bethesda.hu; 4Institute of Genomic Medicine and Rare Disorders, Semmelweis University, 1083 Budapest, Hungary; udvari.szabolcs@semmelweis.hu (S.U.); geresi.adrienn@semmelweis.hu (A.G.); molnar.mariajudit@semmelweis.hu (M.J.M.)

**Keywords:** spinal muscular atrophy (SMA), pilot program, Hungary, newborn screening (NBS)

## Abstract

The growing need to identify spinal muscular atrophy (SMA) patients as early as possible has shifted attention to newborn screening (NBS). The aim of the present study was to evaluate the possibility of including the SMA-NBS in the Hungarian screening panel. As the first step, a government-funded pilot program started in November 2022 and continued until the end of 2023. Evaluation of the first 14 months was followed by the decision to lengthen the program until the end of 2024, which was further supported by the needs of society. Screening tests were performed in both Hungarian national screening laboratories uniformly using the combined EONIS SCID-SMA real-time PCR assay kit by Revvity, for the newborns whose parents gave written consent for the analysis. Altogether, 155,985 newborns were screened during the 26 months of the program, which was 87% of all newborns involved in the national neonatal screens of the same period. All 19 newborns identified on the screen were diagnosed with SMA, confirmed by a multiplex ligation-dependent probe amplification assay (MLPA). The favorable results of the pilot study support the inclusion of the SMA in the national screening panel at the earliest possible date.

## 1. Introduction

The effectiveness of newborn screening programs has been confirmed for various diseases, mainly inherited errors of metabolism. NBS significantly reduces morbidity and mortality and improves outcomes of treatable disorders. The Centers for Disease Control and Prevention (CDC) listed the NBS among the ten greatest public health achievements 2011 [1]. In addition to the gain in health, several studies confirmed the overall cost-effectiveness of NBS [2,3].

Spinal muscular atrophy is an autosomal recessive disease, caused by functional loss of the SMN (survival of motor neuron) protein and resulting in progressive motor neuron death [4]. In ~95% of affected individuals, this loss is due to the homozygous absence of the exon 7 of the *SMN1* gene (OMIM#600354), the remaining 5% are compound heterozygotes with the absence of one *SMN1* allele and a point mutation/small deletion/small insertion. A nearly identical copy of the *SMN1*, the *SMN2* gene can produce only a small amount of functional SMN protein [5]. Thus, *SMN2* modifies the severity of SMA, with higher copy numbers of *SMN2* leading to milder disease presentations [6,7,8]. In the last few years, there was a rapid development in SMA therapies available to prevent symptom development or to slow disease progression: *SMN2* modulators (risdiplam, nusinersen) and *SMN1* gene replacement therapy (onasemnogene abeparvovec) [9]. Clinical data suggest early treatment initiation in children with SMA is critical to achieving optimal outcomes [10]. With therapies available, SMA met the criteria for inclusion in NBS and the Advisory Committee on Heritable Disorders in Newborns and Children added it to the Recommended Uniform Screening Panel in 2018 [11].

Newborn screening for SMA generally uses real-time polymerase chain reaction techniques to assess the patient’s *SMN1* gene, using DNA isolated from dried blood spot (DBS) samples [12]. Many countries started prospective pilot studies for SMA-NBS introduction and in several states, the SMA was thereafter included in the NBS panel [11]. The U.S. Centers for Disease Control and Prevention built quality assurance resources to support SMA-NBS programs by developing reference materials for proficiency testing (applied currently by 115 NBS laboratories) and further on for external quality control.

In Hungary, novel SMA drug therapies started in 2018 with nusinersen [13]. The gene replacement therapy onasemnogene abeparvovec became available in 2019, and risdiplam, an *SMN2* pre-mRNA splicing modifier, was introduced for the early treatment of SMA in 2021.

The rapid implementation of SMA-NBS pilot projects globally has facilitated the collection of precise SMA data, including screening outcome measures, testing time, and treatment [14,15,16,17]. Country-specific birth prevalence data, organizational considerations, and barriers to implementation are important information aimed at achieving the best possible outcome for SMA-NBS locally and globally [12].

Here, we describe the results of the Hungarian SMA pilot study to support the implementation of the screen into the NBS panel.

## 2. Materials and Methods

### 2.1. Initiation and Time Frame of the Pilot Program

The Hungarian SMA NBS pilot program started in November 2022, primarily to examine compatibility with the current NBS system. The government-funded pilot program was originally intended to cover 14 months for neonates born in governmentally funded institutes. The two Hungarian NBS Centres at Semmelweis University, Budapest, and Albert Szent-Györgyi Medical School-University of Szeged participated in the country-wide program. From March 2023 the program was opened to children born in non-governmentally funded birth centers as well. Based on the summary provided at the end of the above period governmental evaluation followed and the primary pilot project was further extended for another year. Families with children born when the program was unavailable due to evaluation (Jan–Feb 2024) could consent to their participation no later than 30 April 2024, upon the program’s restart in March 2024.

### 2.2. Participants of the Program and Sample Collection

DBSs were collected through the ongoing nationwide NBS system, and no extra sample collection was needed to participate in the program. Samples arrived at and were processed in one of the two above-mentioned NBS centers based on the place of birth (the geographic location of analyzed populations according to a consensus-based sampling map covering the whole country). In the Hungarian NBS program, DBS cards are stored at room temperature (for a maximum of 5 years in general).

The Hungarian SMA Centre, Bethesda Children’s Hospital, as the project coordinator, provided all the participants with support and information. Children confirmed positive for SMA were subsequently treated in one of the two SMA centers, Bethesda Hospital or the Pediatric Centre of Semmelweis University.

### 2.3. DNA Isolation and Real-Time PCR Assay

At 48–72 h of age, newborn samples were obtained from heel pricks following standard protocol. DNA was extracted from DBS samples using 3.2 mm punches, in 96-well plates. The Eonis DNA Extraction kit (Revvity, Turku, Finland) was used for manual extractions, according to the manufacturer’s instructions. Real-time PCR assays to detect homozygous absence of the *SMN1* gene exon 7 were used along with detecting the *RPP30* as reference gene as described in the manufacturer’s instructions (Eonis SCID-SMA kit, Revvity, Turku, Finland https://www.revvity.com/product/eonis-platform-eonissystem#product-overview (accessed on 25 March 2025). PCR plates were run on a Quantstudio5/Quantstudio5DX real-time PCR machine (Thermo Fisher, Waltham, MA, USA). Carriers could not be identified in the test. Samples positive in the first tier (no amplification of exon 7 of *SMN1*, but normal presence of *RPP30*) were repeated in duplicate from the same DBS sample within 24 h.

### 2.4. The Diagnostic Genetic Test

Infants with positive results were immediately referred to assigned clinical geneticists providing pre-test genetic counselling for the family, and consent was obtained from parents/guardians for further genetic analysis. Venous blood sampling was performed for the diagnostic test and AAV9 screen. Genetical conformation of SMA and copy number analysis of the *SMN2* gene was performed at the Institute of Genomic Medicine and Rare Disorders Semmelweis University, Faculty of Medicine (Budapest, Hungary). Salsa MLPA probe mix P021-B1 (MRC Holland, Amsterdam, The Netherlands) was used to detect *SMN2* copy numbers.

### 2.5. Ethics Approval

The pilot study was approved by the Medical Research Council Ethics Committee, Hungary by the code of ethical approval 16090-6/2022/EUIG.

### 2.6. Data Analysis

All data were collected in NBS program databases, and calculations were performed with Microsoft Excel. The cut-off cycle threshold (Ct) values determined by the manufacturer of the EONIS kit were applied for the interpretation of results. MLPA results were analyzed using the Coffalyser software v.240129.1959.

## 3. Results

### 3.1. Screening Characteristics: Pilot Study Design

The neuromuscular treatment center of Bethesda Children’s Hospital created the proposal for the study design and started a discussion among participants to optimize the process. The final screening strategy workflow approved by all institutions participating in the program is shown in Figure 1 Before screening, midwiferies and GP Surgeries provided information for the families about the screening program, and informed consent was signed there in advance, or later at birth centers/hospitals (Figure 1). The institutions providing information to parents had to join the program officially by signing a participation consent beforehand.

The Hungarian NBS uses an opt-out approach, testing for all newborns, with the possibility of opting out from screening [18]. Because of the legislation on genetic testing and the strict sensitivity of genetic data, the parents’/legal guardians’ decision to participate in the additional SMA-NBS program was stated in a written form, and an informed consent form was signed, using an opt-in approach in this case.

### 3.2. Screening Outline

From November 2022, during the following 26 months altogether in the two centers, 179,845 newborn DBS samples were tested in the Hungarian NBS program, representing the whole country. The number of neonates tested for SMA was 155,985, an average of 87% of all screening tests performed on newborn samples (Table 1). Families’ attendance constantly increased during the program, as it became more widely known and accepted in time by society and as parents became more familiar with the achievements of the screening program [19].

During the above time, 19 DBS samples were found positive in the multiplex real-time PCR screening assay (Table 2). These results were repeated in duplicate from repeated DBS punches from the same card within a working day (screening laboratories work from Monday to Friday, 8:00 a.m. to 4:00 p.m.). All 19 identified cases were referred to a clinical geneticist physician at the pediatric hospital, where screening was performed, or at the nearest hospital offering pediatric genetic counselling within reach for the family concerned. Clinicians involved in this process were previously consulted with and agreed to align with the principles of the program. After a telephone conversation with the physician, parents were urged to visit the hospital with the child as soon as possible. During the pre-test genetic counselling, parents/guardians were provided information on SMA and potential therapeutic options and the child was examined by a neuropediatrician. Parents were asked to consent to venous blood draw from the newborn for AAV9 titer determination and diagnostic evaluation of SMA. A single specialized center was assigned in the country for this diagnostic evaluation and a concurrent *SMN2* copy number determination. The diagnoses of the 19 SMA patients detected by the first-screen test were all confirmed here and *SMN2* copy numbers were determined by multiplex ligation-dependent probe amplification assay. Upon diagnosis, the child’s parents were contacted again by telephone and referred to the hospital, this time to one of the two assigned treatment centers, where parents were informed about the diagnosis and provided all the necessary information to help their informed decision on the therapy choice.

### 3.3. Screening Timeline

DBS samples were collected following the recommended best practice for NBS (48–72 h) at a median of 3 days after birth (range 2–4). A median of 6.5 days was needed for the sample cards to arrive at the screening centers (range 4–11) mostly by regular postal mail. The median time to have first-tier results was 8.5 days (range 4–21) of life, and the confirmation from repeated DNA punches from the same card was obtained at 9.5 days (range 4–22). Parents were contacted at the child’s median age of 9.5 (range 4–22) and the first visit usually occurred the next business day, at the median age of 10.5 (range 7–23). Families had their diagnostic results at a median age of 14.5 days after their child’s birth (range 9–27) (Table 2) Both laboratories processed samples within 24–48 working hours after DBS card arrival. Nevertheless, occasional delays occurred due to administrative issues causing a shortage of reagents (Table 2 #5 and #6 patients). Data for patients 10, 11, 12 and 13 (Table 2) were not included in the median and range calculations as they opted into the program later, except for patient #10 who was diagnosed after being hospitalized in the period when the program was suspended in Jan-Febr 2024.

### 3.4. Screening Performance

Altogether, 6 children had 2 copies, 11 children had 3 copies, and 3 children had 4 *SMN2* copies from the 19 newborns diagnosed in the screen and 1 diagnosed without NBS during the program period (Table 2). Among the SMA-diagnosed children were 10 males (including one child who was diagnosed when no screening was available) and 10 females.

So far, no false positive screens have been identified in the program, and false-negative test results have not been observed. The screening test’s sensitivity, specificity, and positive predictive value were 100%. The calculated incidence of SMA in the examined period was 1:7799, including 19 SMA patients detected in NBS and the additional patient born during the program stoppage and diagnosed based on symptoms (Table 3).

The retest rate was generally low (~0.1% of all the screened samples). A retest was needed when DNA sample loss during isolation or pipetting errors resulted in real-time PCR failure or inconclusive results. Then, the test was repeated using new punches from the same NBS card.

### 3.5. Distribution of Identified Patients Across Hungary

Although the SMA newborn screening program was announced as a nationwide program, the participation of the families was partly limited by the birth institutions’ intention to join. As awareness was raised, main hospitals from all regions enrolled the program.

Based on Eurostat data [20], the designated predominantly urban and rural regions were compared in the country, and the residency of the SMA-diagnosed patients’ families during the program was depicted by regions (Figure 2A). According to the regional map of Hungary, the rural population accounts for 18% of the whole population, meanwhile 35% of the SMA-diagnosed children are inhabitants of a rural region. As for the degree of urbanization, the Eurostat-recommended classification was followed and we used the population size threshold above 50,000 inhabitants for urban centers, below 5000 inhabitants for rural areas and between 5000 and 50,000 people for (predominantly) urban or intermediate areas (in Hungary it also involves large villages). Exact calculations were based on official regional population data regularly published on the Hungarian Central Statistical Office website for all the regions’ settlement types [21]. Based on these calculations (Figure 2B), a similar relationship was found: 25% for the whole population and 45% for the SMA-diagnosed children’s families’ distribution in rural regions.

As for various levels of regional divisions of the country, data are illustrated in Figure 2C,D, for large geographic regions and the three major geographic regions of the country, respectively. Certain parts show a greater incidence of children born with SMA, predominantly in the Great Plain and North of the country and counties, e.g., Szabolcs-Szatmár-Bereg. However, counties in other regions (e.g., Veszprém, Somogy in the Transdanubia), and the capital city in Central Hungary, where 17.6% of the country population is concentrated [21] (1 January 2024 data) are greatly underrepresented.

## 4. Discussion

The earliest possible detection of SMA is considered one of the most important factors in effectively treating the disease [22]. A newborn screening system can provide the best opportunity for presymptomatic testing and the best chance of successful therapy [10]. Current NBS strategies and practices vary across European countries; however, the inclusion of SMA in the national screening programs seems to be a reasonable goal in the region and a demonstrated intention of the national stakeholders [23,24]. Moreover, several real-world data support the cost-effectiveness of SMA NBS screening, when combined with early-stage treatment [25,26].

As treatment options widened and became more acknowledged by society, a growing demand from the side of parents and patient organizations prompted the development of the Hungarian SMA national pilot program. Bethesda Children’s Hospital is the pioneer SMA treatment center in Central Eastern Europe, where SMA was first treated successfully with gene therapy and was an expert candidate to conduct the program. They informed all maternity hospitals and the involved health professionals; information pamphlets and consent documents were made available on the website of Bethesda Hospital. Dedicated contact details were provided to parents as well. Because of the great media coverage of SMA-related information, they organized campaigns in written and social media to convey accurate information and encourage participation in screening from both the side of parents/guardians and birth institutions.

SMA screening represented the first NBS program in Hungary using molecular DNA-based methodology, which posed several challenges [27] to be solved by the screening laboratories. As early detection was vital, samples had to be processed at the designated screening centers, where regular screening was also performed. First, both screening centers established suitable new facilities, partly separated from the existing laboratory units to avoid any possible sources of contamination and equipped them with the proper genetic testing platforms. Being among the most widely used and approved screening tests, both laboratories decided to use the Eonis system of Revvity [28]. In addition to the kit’s internal controls, the assay’s analytical performance was regularly monitored through participation in the CDC’s Newborn Screening Quality Assurance Program for SMA.

Moreover, the integration of genetic screening into the operating NBS system (even in the pilot stage) required a different approach, because of the current legislation on the protection of human genetic data. To opt-in for the SMA screening, a consent form had to be signed by parents/guardians at the local midwives/medical doctors beforehand or in the birth center after being informed about SMA screening. In addition, general data protection sheets related to the sample treatment were handed over and had to be signed by the parents. The whole process emerged as an extra administrative burden for the hospitals/GP surgeries involved, and the lack of human resources was a problem in trying to increase the number of participants in the program. Consequently, some parents did not gain information on the possibility of the screening or were informed only well after the delivery from other sources. In such instances, SMA screening was performed based on a case-by-case decision additionally upon request, using the newborn’s previously arrived card. Although regular NBS was accessible for children born at home or private birth centers/hospitals, cost-free availability of the SMA test was limited to governmentally funded institutes in the first 4 months of the program, resulting in a lower participation rate at the beginning. Altogether, better communication approaches raised the awareness of the participants and parents, and the governmental involvement of the private birth centers also prompted a higher screening coverage of the newborn population. The participation rate in the SMA pilot screening rose from the starting point of 40–45% to approximately 80–90% [19]. Despite the available information sheets and the possibility of contact with healthcare providers, there was a common misunderstanding from the side of the parents and other participants. Although it was clearly stated that screening centers did not have the legal right to communicate screening results to parents, a constant, apprehensible need persisted from their side. Because of the genetic information content of the screen, the acting law allows the results to be communicated only in the frame of genetic counselling by a clinical geneticist, and this opportunity was offered automatically only for parents whose child has had a positive result in SMA NBS.

Occasional fluctuations in the screening timeline are caused partly by the imbalanced nature of postal delivery which is also sustained by a lag in card posting. Only hospitals with a greater birth number have the opportunity to use courier service. First-tier screening results were unfavorably delayed in a few cases by a shortage of the reagents due to administrative issues (Table 2). However, the laboratory standards allowed the completion of the screen within 1–2 working days after arrival. The pilot program was temporarily ceased for two months in Jan/Feb 2024 for administrative reasons. Upon resumption of the screening parents/guardians had the right to request additional SMA screening using the original card of the newborn until the end of April 2024, and screening centers processed these additional requests at the earliest possible time. From March 2024 the program restarted, and samples continued to be analyzed as previously, along with the additional requests arriving by the end of April. This temporary suspension caused a significant delay in the SMA screens and children born during this period with SMA were diagnosed at 2–3 months of age (Table 2 #11, 12 and 13). One child (Table 2 #10) was born in early 2024 and was hospitalized, diagnosed and treated post-symptomatically. The accuracy outcomes of the SMA pilot in Hungary were in good correlation with published pilot experiments for sensitivity, specificity, and positive/negative predictive values (Table 3) [12].

Our data presented here show a nearly two times higher incidence of SMA in predominantly rural than urbanized regions (Figure 2A,B). Moreover, certain geographical areas showed a relationship to the occurrence of the disease (Figure 2C,D), which resulted in largely unbalanced hit numbers between the two screening centers (Table 1). This might be due to actual emerging regional differences or else, the results may turn out to be different in a longer period. A possible explanation might be a greater genetic risk of certain subpopulations due to consanguineous marriages with less likelihood of the residents’ propensity to move, in these predominantly rural areas. Or else, there might be some underrepresented cohorts in testing for any reason.

Many countries offer DNA-based screening tests for newborns for severe combined immunodeficiency (SCID) [29,30,31]. This test can easily be multiplexed with an SMA screen [32]. No other special reagents or commercial kits would be required to screen SCID in Hungary in the future, as the Eonis kit used in our screen can simultaneously quantify T-cell receptor excision circles (TRECs) from the same DNA sample and the same reaction [33,34].

## 5. Conclusions

The feasibility of an SMA large-scale screening program was successfully tested in practice and real-life efficiency was confirmed in the newborn population [23]. SMA diagnoses are now recognized as clinical emergencies as early therapy ensures the best efficacy [10]. Positive screening test results were reported to healthcare professionals before the child’s median age of 10.5 days in our pilot project between November 2022 and December 2024. Immediate clinical follow-ups were provided for the children screened and diagnosed with SMA, followed by suitable therapy. Government decision making is in progress to help legal framing evaluate the future expansion of core newborn screening with the involvement of a genetic test for SMA. Screening centers are ready to step to the next level, and the implementation of the SMA pilot project into regular NBS is strongly encouraged not only from the side of healthcare providers but also from the side of society.

## Figures and Tables

**Figure 1 IJNS-11-00029-f001:**
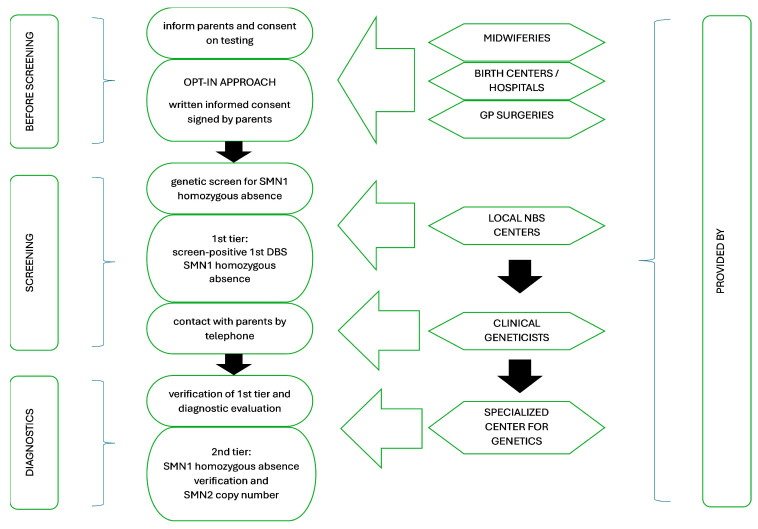
The pathway from screening to diagnosis of the screen-positive newborns in the Hungarian SMA pilot program.

**Figure 2 IJNS-11-00029-f002:**
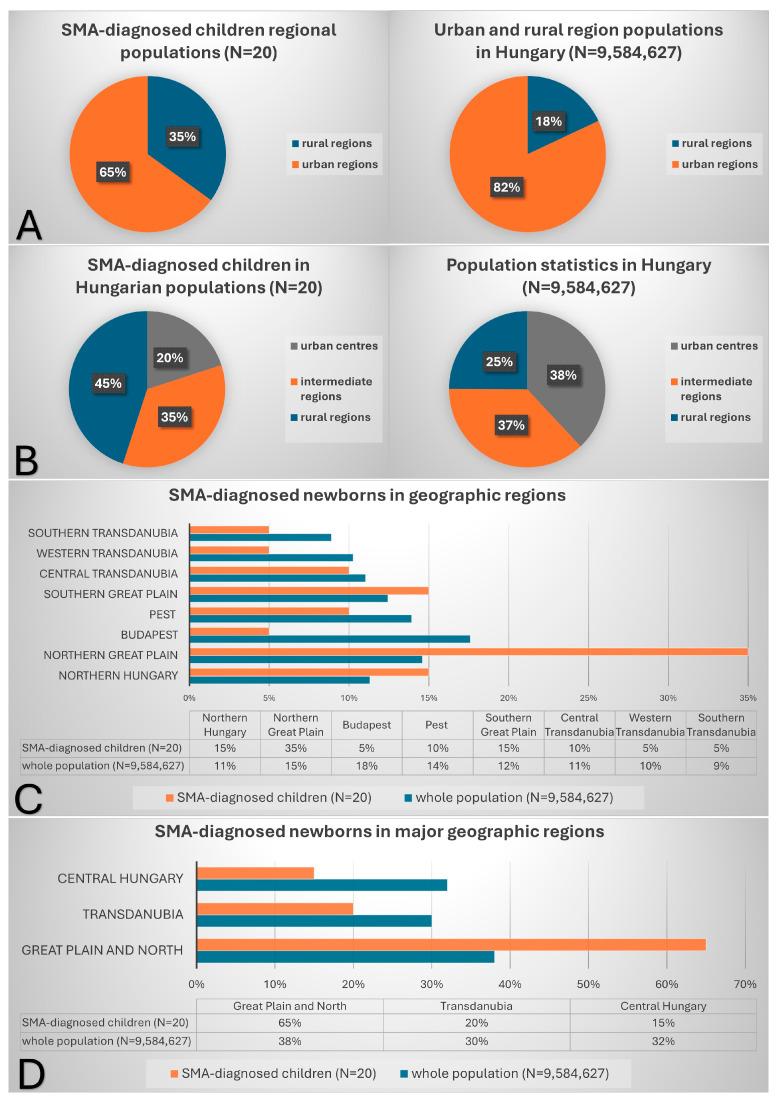
Regional distribution of SMA patients diagnosed during the program. The number of patients percentage or inhabitants percentage of the total is indicated in each part. Population data are from the official statistical website of Hungary (www.ksh.hu) updated on 1 January 2024. The SMA-diagnosed children were found during the time frame of the program as newborns. (**A**) Eurostat defined urban and rural area population distributions. (**B**) Population data based on the population size threshold above 50,000 inhabitants for urban centers, below 5000 inhabitants for rural areas and between 5000 and 50,000 people for intermediate areas, calculated based on the exact population size of the SMA-diagnosed child’s residency data and the detailed regional settlement population data of the country. (**C**,**D**) SMA-diagnosed newborns compared to inhabitants at multiple geographic regional area levels.

**Table 1 IJNS-11-00029-t001:** Neonates screened within the pilot study from 1 November 2022 to 31 December 2024 in the two screening centers. One child was diagnosed with SMA based on the symptoms when the pilot program was not available.

SMA Pilot Study from November 2022 to December 2024						
	Number of all neonates tested for SMA in		Number of all neonates tested in all NBS programs in		Number of SMA-diagnosed neonates in the program	
	Budapest	Szeged	Budapest	Szeged	Budapest	Szeged
	84,670	71,315	96,456	83,389	6	13
**summary**	**155,985 (87% of all NBS)**		**179,845**		**19 + 1**	

**Table 2 IJNS-11-00029-t002:** Diagnostic timeline in post-natal days for the 19 SMA patients identified in the pilot screen (patient #10 was diagnosed symptomatically with SMA during a temporary suspension of the SMA pilot study in early 2024) in the order of their birth date. Result data for children #10, 11, 12, and 13 (in italics) were not included in the median and range values. Patients #11, 12, and 13 were tested retrospectively (see also Notes).

ID#	DBS Sampling (Age in Days)	DBS Received by NBS Centre (Age in Days)	First-Tier Result (Age in Days)	First-Tier Result Repeated (Age in Days)	Parents Contacted (Referral to Specialist) (Age in Days)	First Visit (Specialist Review) (Age in Days)	Second-Tier Result (Diagnostic Result) (Age in Days)	*SMN2* Copy Numbers	Notes
1	3	5	6	7	7	8	11	3	
2	3	7	7	8	8	9	12	3	
3	3	8	8	9	9	10	15	2	
4	3	6	6	7	7	8	12	3	
5	3	6	14	15	15	18	22	4	shortage of PCR reagents ^1^
6	3	8	21	22	22	23	27	3	shortage of PCR reagents ^1^
7	2	4	4	4	4	7	9	2	
8	4	5	5	5	8	9	11	2	
9	2	10	10	11	11	12	16	3	
*10*	2	6	*61*	*61*	*-*	*-*	*50*	2	no NBS for SMA available ^2^
*11*	2	7	*111*	*111*	*111*	*112*	*118*	3	no NBS for SMA available ^3^
*12*	3	6	*109*	*109*	*109*	*110*	*113*	4	no NBS for SMA available ^3^
*13*	3	8	*64*	*65*	*65*	*68*	*70*	4	no NBS for SMA available ^3^
*14*	3	5	6	9	9	10	18	3	
15	3	6	11	12	12	13	16	3	
16	2	8	9	10	10	11	12	3	
17	2	6	8	8	8	9	12	3	
18	3	7	11	12	12	13	14	2	
19	3	10	11	12	12	15	16	3	
20	2	11	12	13	13	14	15	2	
median	3.0	6.5	8.5	9.5	9.5	10.5	14.5		
range	2–4	4–11	4–21	4–22	4–22	7–23	9–27		

Notes: ^1^: Due to financial reasons, there was a delay in purchasing reagents; ^2^: Patient #10 was identified based on symptoms during the program suspension, and DBS was tested after SMA was diagnosed in the hospital; ^3^: DBS for the patient #11, #12 and #13 were tested upon program restart after a temporary 2-months stop early 2024.

**Table 3 IJNS-11-00029-t003:** Performance data of the SMA pilot screen.

Statistical Characterization of the Hungarian SMA Pilot Program from November 2022 to December 2024	
Number of infants screened for SMA	155,985
Number of infants with positive tier 1 test among infants screened (% of infants screened)	19 (0.0122%)
Number of infants with a confirmed SMA diagnosis among infants screened	19
Number of infants with a confirmed SMA diagnosis among infants not screened	1
Incidence of newly diagnosed SMA in the whole newborn population (screened and unscreened)	1:7799
Sensitivity	100%
Specificity	100%
Positive predictive value	100%

## Data Availability

All research data can be found in the Hungarian NBS database.

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
