# Peer review of "Results of the Hungarian Newborn Screening Pilot Program for Spinal Muscular Atrophy"

_2409-515X, 2025, doi:10.3390/ijns11020029_

Round 1
Reviewer 1 Report
Comments and Suggestions for Authors
In this manuscript, the authors present the results of the NBS pilot for SMA in Hungary, adding to the literature of other states and countries on pilot and post-NBS implementation data. I have the following specific comments/suggestions for consideration by the authors:
- Please italicize all gene names
- More clarity on the program suspension would be helpful to understand if this was a situation unlikely to recur post-implementation. This will help me determine whether it is appropriate to exclude patients 10-13 in the median age of call-out calculations.
- Is it to be implied that pt #13 did not present with symptoms prior to diagnostic result at 70 DOL?
- Can authors provide additional detail on whether all infants screened/confirmed were treated? Early diagnosis is most meaningful when there is robust access to treatment. The conclusion sentence "Immediate clinical follow-ups were provided for the children screened and diagnosed with SMA, followed by suitable therapy" suggests that all babies through this pilot were treated. Please confirm this is the correct interpretation of this statement.
- Re: Figure 2, were all the Hungarian counties in C part of the NBS program? If not, I am not sure this particular figure is appropriate/necessary. Even if so, I think this figure could be more concisely represented.
Reviewer 2 Report
Comments and Suggestions for Authors
Hegedűs et al. presented the results of a pilot program for spinal muscular atrophy newborn screening (SMA-NBS) conducted in Hungary in the period of 2022-2024. According to the manuscript, newborns whose parents gave written consent for the analysis were uniformly screened using Revvity's EONIS SCID-SMA real-time PCR combined assay kit. A total of 155,985 newborns were screened during the 26-month program. All 19 newborns identified by screening were diagnosed with SMA and confirmed by multiplex ligation-dependent probe amplification (MLPA). The manuscript also described a total of 20 cases during this period, including one case discovered by clinical symptoms during the program period. The authors concluded that "the positive results of the pilot study support the inclusion of SMA in national screening panels as early as possible."
This manuscript is very appealing because of the authors' candid explanation. The authors honestly report the problems they faced in implementing SMA-NBS, which will be useful for researchers in countries working on implementing SMA-NBS in the future. Table 2 provides examples of cases where there was a lack of funds to provide PCR reagents, screening was delayed due to program interruptions, and diagnostic results arrived before screening results. The authors also pointed out that there are limited human resources involved in SMA-NBS (line 288) and that the free service is not available for home births or private hospital births (line 294). They also pointed out that occasional fluctuations in screening timelines are largely due to delays in mailing cards (line 309). According to the authors, only hospitals with high birth volumes have access to home delivery services.
Below are some suggestions for minor revisions to improve this manuscript.
(1) Throughout the paper, it would be better to show gene names in italics.
(2) In the Methods section, why not cite the URL of the EONIS SCID-SMA real-time PCR assay kit website to get information about this product?
(3) In Table 2, it would be better to add (age in days) to the headings of all items other than the SMN2 gene copy number.
(4) In Table 3, it would be better to write PPV as positive predictive value.
(5) The copied and pasted image in Figure 2 is not clear. It should be replaced by clearer one.
(6) In Figure 2, values are expressed as decimals in A and as percentages in B. It would be better to use the same notation.
(7) In Figure 2, the font is too small and difficult to read in C, D, and E. Perhaps it would be better to enlarge it.
Reviewer 3 Report
Comments and Suggestions for Authors
I appreciate the opportunity to review this important paper on the implementation of newborn screening (NBS) for spinal muscular atrophy (SMA) in Hungary. This is a timely and critical subject, with substantial implications both locally and globally. The availability of effective treatments in Hungary underscores the importance of early detection and intervention. While the manuscript provides valuable insights into the outcomes of the screening program, I would like to offer a few minor suggestions to improve the paper.
1. Terminology in Figure 1:
I recommend that the authors use the term "absence" instead of "deletion" in Figure 1, as the SMN1 gene can be absent due to either a deletion or gene conversion to SMN2.
2. Clarification on qPCR Terminology:
I suggest a minor correction regarding the use of "qPCR". Technically, "qPCR" refers to quantitative real-time PCR, whereas the authors describe a qualitative real-time PCR method (presence/absence detection). The correct term for the methodology used would therefore be "real-time PCR." While "qPCR" is commonly used in the literature, precision in terminology would enhance the scientific rigor of the manuscript, but I do not insist that the authors correct this.
3. Additional Details on the Screening Program:
To further improve the manuscript, I would encourage the authors to provide additional details regarding the screening process. Specifically, information on the following points would be valuable:
-
The storage conditions (e.g., room temperature, -20°C, etc.) and storage duration (how many years) of the dried blood spot (DBS) cards.
-
I am curious if there were any cases where a blood transfusion was required, and if so, whether the authors believe that transfusions could interfere with the screening results. If transfusions did occur, was the sample taken prior to the transfusion, or was it potentially repeated after the transfusion? If repeated, how long after the transfusion was the sample taken?
-
An overview of how treatment decisions are made in Hungary for SMA patients: Who determines the most optimal treatment, and what criteria are used? Is it based on parental preference, clinical opinion, or a consensus between the two? Additionally, a discussion on how clinicians decide between the three available treatment options would add valuable context to the paper.
In conclusion, this paper makes a significant contribution to the field of SMA newborn screening. I would like to offer my best wishes to the authors as they continue to advance the implementation of newborn screening in Hungary. I hope that any future reagent shortages, as experienced during the pilot phase, will be avoided, ensuring that all SMA patients benefit from timely and accurate diagnosis and treatment.
